

# In silico screening of chalcones and flavonoids as potential inhibitors against yellow head virus 3C-like protease

Kanpong Boonthaworn[1,2], Kowit Hengphasatporn[3], Yasuteru Shigeta[3], Warinthorn Chavasiri[4], Thanyada Rungrotmongkol[5] and Puey Ounjai[1,2]

[1] Department of Biology, Faculty of Science, Mahidol University, Ratchathewi, Bangkok, Thailand
[2] Center of Excellence on Environmental Health and Toxicology, Ministry of Education, Ratchathewi, Bangkok, Thailand
[3] Center of Computational Sciences, University of Tsukuba, Tsukuba, Ibaraki, Japan
[4] Department of Chemistry, Faculty of Science, Center of Excellence in Natural Products Chemistry, Chulalongkorn University, Pathum Wan, Bangkok, Thailand
[5] Structural and Computational Biology Research Unit, Department of Biochemistry, Faculty of Science, Chulalongkorn University, Pathum Wan, Bangkok, Thailand

Corresponding authors
Thanyada Rungrotmongkol,
thanyada.r@chula.ac.th
Puey Ounjai, puey.oun@mahidol.edu

## ABSTRACT

Yellow head virus (YHV) is one of the most important pathogens in prawn cultivation. The outbreak of YHV could potentially result in collapses in aquaculture industries. Although a flurry of development has been made in searching for preventive and therapeutic approaches against YHV, there is still no effective therapy available in the market. Previously, computational screening has suggested a few cancer drugs to be used as YHV protease (3CL$^{pro}$) inhibitors. However, their toxic nature is still of concern. Here, we exploited various computational approaches, such as deep learning-based structural modeling, molecular docking, pharmacological prediction, and molecular dynamics simulation, to search for potential YHV 3CL$^{pro}$ inhibitors. A total of 272 chalcones and flavonoids were in silico screened using molecular docking. The bioavailability, toxicity, and specifically drug-likeness of hits were predicted. Among the hits, molecular dynamics simulation and trajectory analysis were performed to scrutinize the compounds with high binding affinity. Herein, the four selected compounds including chalcones cpd26, cpd31 and cpd50, and a flavonoid DN071_f could be novel potent compounds to prevent YHV and GAV propagation in shrimp. The molecular mechanism at the atomistic level is also enclosed that can be used to further antiviral development.

# INTRODUCTION

Yellow head virus (YHV) is one of the major pathogens in shrimp aquaculture, causing catastrophic loss to aquacultural industries across South-East Asian countries (*Stentiford, Bonami & Alday-Sanz, 2009*). YHV is one genotype in a complex of closely relative viruses, including Gill-associated virus (GAV), a severe shrimp pathogen found in Australia

(*Ziebuhr et al., 2003*). As the name suggested, the most recognizable sign of infection is the appearance of yellowish coloration on the cephalothorax (*Chantanachookin et al., 1993*). The virus is well known for its virulence as it can slay the entire farm within 2–3 days of emergence (*Assavalapsakul, Smith & Panyim, 2006*). Several methods have been developed against YHV infection, such as the feeding of herbal extract from guava (*Direkbusarakom et al., 1997*), using rat raised poly-clonal antibody against YHV spike protein (*Chaivisuthangkura et al., 2008*), using RNA interference technique against YHV RNA polymerase (*Tirasophon, Roshorm & Panyim, 2005*; *Yodmuang et al., 2006*), helicase (*Tirasophon, Roshorm & Panyim, 2005*; *Yodmuang et al., 2006*), protease (*Assavalapsakul, Chinnirunvong & Panyim, 2009*; *Tirasophon, Roshorm & Panyim, 2005*; *Tirasophon et al., 2007*; *Yodmuang et al., 2006*), and selective breeding (*Moss et al., 2012*). However, it is still insufficient to prevent massive losses caused by YHV outbreaks. The development of effective antiviral agents against the infection is still required.

Several antiviral drugs targeting the viral replication process have been developed. These drugs often interrupt the orchestrated action of the viral nonstructural proteins (*Dinesh et al., 2020*). Viral protease is one of the most exciting targets for antiviral drug development due to its structural simplicity, small size, and function in producing other enzymes and structural proteins necessary to produce mature virions (*Dinesh et al., 2020*). Chymotrypsin-like proteinase (3CL$^{pro}$) is a protease encoded by most viruses in Nidovirales, including YHV. 3CL$^{pro}$ plays a pivotal role in the viral replication process by cleaving the viral polyproteins at multiple conserved sites to release several vital replicative proteins (*Ziebuhr et al., 2003*). Due to the unavailability of the 3D structure from X-ray crystallography, nuclear magnetic resonance (NMR), or cryogenic electron microscopy (cryo-EM) of YHV 3CL$^{pro}$, homology-based modeling has been conducted to gain the structural basis of this viral protease (*Unajak et al., 2014*). The YHV 3CL$^{pro}$ model was used in virtual screening for potential compounds from the cancer therapeutic drugs database. NSC122819 was the best promising candidate as it showed relatively good results in an inhibition experiment against bacterially expressed YHV 3CL$^{pro}$. However, this compound is a chemotherapeutic drug derivative to podophyllotoxin, causing a cytotoxic effect and inhibiting topoisomerase II activity (*Dombernowsky, Nissen & Larsen, 1972*). Although the potential compound exhibited a good result the in vitro experiments, it is still challenging further to develop NSC122819 for exploitation in the farm setting. Therefore, a search for an alternative compound with higher bioavailability suitable for the treatment of YHV infection would be of great interest.

Flavonoids, the common polyphenols produced by plants, are the most abundant polyphenol in animal diets (*Dai & Mumper, 2010*). As secondary plant metabolites, which are not essential for the living of the plant, flavonoids are responsible for various biological activities, such as plant coloration (*Yoshida, Oyama & Kondo, 2012*), UV filtration (*Takahashi & Ohnishi, 2004*), molecular signaling (*Brunetti et al., 2018*), detoxifying (*Samanta, Das & Das, 2011*), and function as anti-microbial agents (*Al Aboody & Mickymaray, 2020*). Moreover, flavonoids also showed several pharmaceutical properties (*Agrawal, 2011*). Due to their low toxicity and availability, flavonoids are a well-known component of herbal medicines and even dietary supplements (*Kumar & Pandey, 2013*).

Chalcones are precursors of flavonoids and isoflavonoids. The compounds are one the most widespread natural products found in various plants. Chalcones also considered as a subset of flavonoids since their chemical structures consist of open-chain flavonoids as the two aromatic rings joined together with unsaturated carbonyl of three carbon atoms (*Sahu et al., 2012*). Chalcones also showed a broad spectrum of bioactivities, same as flavonoids (*Elias, Beazely & Kandepu, 1999*). Flavonoid and chalcones have been reported to be inhibitor of several virus proteases, such as HIV-1 protease (*Turkovic et al., 2020*; *Xu et al., 2000*), dengue-2 protease (*Hengphasatporn et al., 2020*; *Kiat et al., 2006*; *Srivarangkul et al., 2018*), MERS-CoV 3CL^pro (*Jo et al., 2019*), and SARs-CoV 3CL^pro (*Hengphasatporn et al., 2022*; *Nguyen et al., 2012*; *Park et al., 2016*). In recent studies, as a presence of COVID-19 incident, line of evidence suggested that flavonoids and chalcones could have a potent inhibitory effect against SARS-CoV-2 3CL^pro (*Da Silva et al., 2020*; *Das et al., 2021*; *Jo et al., 2020*; *Vijayakumar et al., 2020*).

According to previous studies about potentials of flavonoid and chalcone as viral protease inhibitors, as well as their low toxicity and bioavailable nature, a set of flavonoids and chalcones from our database was considered for identification of bioactive drug candidates for inhibition of YHV major protease, YHV 3CL^pro, using computational approaches (*Abudayah et al., 2022*; *Al-Sha'er, Basheer & Taha, 2023*). The YHV 3CL^pro model predicted by deep learning-based structural modeling was used to identify potential candidates of chalcones and flavonoids by molecular docking, drug-likeness prediction, and molecular dynamics (MD) simulations. The insights offered from this work can thus open a new avenue to exploit both chalcones and flavonoids in preventive and therapeutic approaches against YHV infection. Since YHV and GAV are closely related, the structure of GAV is also included in the screening process to obtain the candidate compounds that can be antiviral agents for both pathogens.

## MATERIALS & METHODS

### 3CL^pro structure modeling

Due to lack of the 3D structure available, the 3D model of 3CL^pro of YHV and GAV were constructed based on their amino sequences. The amino acid sequence of YHV 3CL^pro was obtained from the NCBI protein database (NCBI reference sequence: ACH99403.1, accessed in October 2021). The amino sequence of GAV 3CL^pro was extracted from ORF1, a polyprotein pp1a, available on NCBI database (NCBI reference sequence: YP_001661453.1). The two proteins were modelled using ColabFold web software (*Mirdita et al., 2022*) with AlphaFold v.2.0 (*Jumper et al., 2021*). Subsequently, the protonation state of ionizable residues was assigned at pH 7.4 using the Open Babel (*O'Boyle et al., 2011*). The catalytic histidine, H63, was in neutral form with protonated $\delta$ nitrogen. The side chain of catalytic dyad H63 and C152 was adjusted to be in active conformation before energy minimization using Chimera software (*Huang et al., 1996*). The model quality score and local distance difference test (lDDT) were used to determine the local quality of the models, while the regions with lDDT value <50 were manually trimmed out from the model.

## Molecular docking

According to previous works about potentials of flavonoid and chalcone as viral protease inhibitors , HIV-1 protease (*Turkovic et al., 2020*; *Xu et al., 2000*), dengue-2 protease (*Hengphasatporn et al., 2020*; *Kiat et al., 2006*; *Srivarangkul et al., 2018*), MERS-CoV 3CL$^{pro}$ (*Jo et al., 2019*), and SARs-CoV 3CL$^{pro}$ (*Hengphasatporn et al., 2022*; *Nguyen et al., 2012*; *Park et al., 2016*), as well as their low toxicity and bioavailable nature, the set of 223 chalcones and 49 flavonoids from in-house database (*Sangpheak et al., 2019*) was considered for identification of bioactive drug candidates for inhibition of YHV major protease, YHV 3CL$^{pro}$, in this study. The structure of the reported anti-YHV 3CL $^{pro}$ agent, NSC122819 (*Unajak et al., 2014*), was downloaded from the ZINC database. The 2D structures of all studied compounds were depicted in supplemental Fig. S1. Each compound from in-house database and the known potent compound NSC122819 was docked into the 20x20x20 Å$^3$ box centered at the center of 3CL$^{pro}$ catalytic residues, H63 and C152, using AutoDock VinaXB (*Koebel et al., 2016*) with 100 runs. Among the different configurations, the ligand conformation with the highest binding affinity was selected for further analysis. To validate reliability of molecular docking result, the known inhibitor NSC122819 was used as a template to generate 50 decoy molecules with Directory of Useful Decoys, Enhanced (DUD-E) sever (*Mysinger et al., 2012*). The decoys and candidate molecules went through the molecular docking process again to generate receiver operating characteristic (ROC) curve (*Empereur-Mot et al., 2015*).

## Drug-likeness prediction

The screened compounds from the molecular docking study were submitted to SwissADME (http://www.swissadme.ch/) (*Daina, Michielin & Zoete, 2017*) to calculate physicochemical properties. The molecular weight (MW), the numbers of hydrogen bond donors (HBD) and acceptors (HBA), rotatable bond (RB), polar surface area (PSA), and lipophilicity (LogP) were used to predict the compound drug-likeness following criteria of Lipinski's rule of five which are: (i) Mw $\leq$ 500 Da, (ii) HBD $\leq$ 5 and HBA $\leq$ 10, (iii) RB $\leq$ 10, (iv) PSA $\leq$ 140 Å and (v) LogP $\leq$ 5 (*Lipinski, 2004*). The molecule which breaks two or more criteria is considered a non-drug-like compound. The Brain Or IntestinaL EstimateD permeation method (BOILED-Egg) (*Daina & Zoete, 2016*) was also obtained from SwissADME.

## Molecular dynamics simulation

The docked ligands/YHV 3CL$^{pro}$ complexes were performed by 200-ns MD simulations in periodic boundary conditions using AMBER20 (*Case et al., 2021*). The ligand structures were optimized by the HF/6–31G(d) level of theory using Gaussian 09 software (*Frisch et al., 2009*), and the electrostatic potential (ESP) charges were generated at the same method (*Sanachai et al., 2020*). Subsequently, the restrained ESP (RESP) charges of ligand were obtained using the ANTECHAMBER module. The protein was treated by AMBER ff16SB forcefield, while the ligand parameters were retrieved from generalized AMBER force field version 2 (GAFF2). The TIP3P water molecules were added to solvate the complex in a cubic box with a dimension extended at least 13 Å from the system surface. Counter ions, Na+, were added to neutralize the system. To diminish the unfavorable

contacts and steric hindrances, the added waters and ions were minimized using steepest descent (SD) and conjugated gradient (CG) methods for 1500 iterations with restrained protein-ligand complex using a force constant of 500 kcal/mol $\text{Å}^2$, followed by the same minimization procedure with 1000 iterations for the whole system. The SHAKE algorithm was applied to constrain all hydrogen atoms connected with covalent bonds (*Ryckaert, Ciccotti & Berendsen, 1977*). The particle-mesh of Ewald's summation method (*York, Darden & Pedersen, 1993*) was used to treat the long-range electrostatic interactions with 10 Å cut-off distance. Temperature and pressure were controlled by the Langevin dynamics (*Welling & Teh, 2011*) and Berendsen barostat (*Berendsen et al., 1984*), respectively. The time step of 2 fs was used. Two steps of 500-ps MD simulations were carried out on each system with a restrained position of the ligand/protein complex by a force constant of 5.0 and 2.0 kcal/mol $\text{Å}^2$, respectively. Then, MD simulations of ligand/protein complexes without any restraint were continued up to 200 ns. The trajectories extracted every 10 ps were analyzed in terms of root mean square displacement (RMSD), number of contact atoms and number of H-bonds between protein and ligand with CPPTRAJ module (*Roe & Cheatham III, 2013*).

To calculate the binding affinity of the interaction between candidate compounds and 3CL$^{\text{pro}}$, the Molecular Mechanics/Poisson–Boltzmann Surface Area (MM/PBSA) and the Molecular Mechanics/Generalized Born Surface Area (MM/GBSA) method were performed using 100 snapshots from the last 50 ns of the simulations. The internal dielectric constant was set to be 1. The external dielectric constant was set to 80. The surface tension was set to 0.0072 kcal/mol $\text{Å}^2$. And the solvent probe radius was set to 0.14 Å. Moreover, the solvated interaction energy (SIE) approach was also adopted to evaluate the binding affinity of ligand toward 3CL$^{\text{pro}}$ (*Naïm et al., 2007*). The compounds with the same level or higher binding affinity against YHV 3CL$^{\text{pro}}$ compared to the known inhibitor NSC122819 were selected for investigation on intermolecular interactions with protein target, i.e., MM/GBSA per-residue decomposition free energy (*Miller III et al., 2012*), with cutoff at $\Delta G_{\text{bind}} = -1$ kcal/mol. The top molecules with highest binding affinity were simulated again along with the known inhibitor to confirm repeatability of the simulation.

## RESULTS AND DISCUSSION

### Virtual screening

The 3D structures of YHV and GAV 3CL$^{\text{pro}}$ were modeled using the CollabFold web application (*Mirdita et al., 2022*) with AlphaFold v.2.0 and MMseqs2 (*Jumper et al., 2021*; *Mirdita et al., 2021*) according to the amino acid sequences obtained from the NCBI database (codes: ACH99403.1 and YP_001661453.1). The two models of YHV 3CL$^{\text{pro}}$, YHV_M1 and YHV_M2, and a model of GAV 3CL$^{\text{pro}}$ with the best model quality score are chosen and given in supplemental Fig. S1. The catalytic dyad H63 and C152 are identified in both YHV and GAV 3CL$^{\text{pro}}$, as described in the previous study of GAV (*Ziebuhr et al., 2003*). GAV and YHV are closely related both genetically and morphologically (*Chantanachookin et al., 1993*; *Cowley et al., 2000*; *Ziebuhr et al., 2003*). The binding pockets of the YHV and GAV models are considered and compared in Fig. 1A. It can be seen that the binding

pockets of YHV_M1, YHV_M2, and GAV 3CL $^{pro}$ models were well-aligned, and the amino acid sequence at the binding site was highly conserved (92.6% identity, Fig. S2). In Fig. 1B, the model YHV_M2 exhibited a higher z-score (11.8) to the GAV binding pocket than YHV_M1 (11.2). The 3CL$^{pro}$amino sequences of GAV and YHV are highly conserved with 94% similarity (Fig. S2). Thus, the model YHV_M2 was chosen to be a representative model for YHV 3CL$^{pro}$ in the following studies. The previous homology based model of YHV 3CL$^{pro}$ suggested H30, D70, and C152 as 3 catalytic residues that form the catalytic triad of the protease (*Unajak et al., 2014*). On the contrary, our predicted model from AlphaFold2 indicated that the YHV 3CL$^{pro}$ employs H-C catalytic dyad for proteolytic mechanism which is in good agreement with the previous experimental data obtained from GAV 3CL $^{pro}$ (*Ziebuhr et al., 2003*). Considering the genetic similarity and mactching residue position of 3CL$^{pro}$ H63 and C152 in the catalytic dyad of both GAV and YHV 3CL$^{pro}$ (Fig. 1A), we identified H63 as one of the catalytic residues in our model instead of the previously proposed H30 (*Unajak et al., 2014*). The designation of H63 as part of the YHV 3CL$^{pro}$ catalytic dyad is in great accordance with the previous experimental reports (*Ziebuhr et al., 2003*).

Molecular docking with 100 runs was employed to generate the possible ligand binding configurations in the binding pocket of YHV and GAV 3CL$^{pro}$. For each compound, the best configuration with lowest binding free energy ($\Delta G$) was chosen to be ranked and compared with the known YHV inhibitor, NSC122819 (*Unajak et al., 2014*). The $\Delta G$ of NSC122819 was of $-8.0$ kcal/mol in YHV 3CL$^{pro}$ and $-6.7$ kcal/mol in GAV 3CL $^{pro}$. Among the 272 studied compounds (Fig. 1C), there were 111 compounds with $\Delta G$ lower than that of NSC122819. In this study, only the top 15% consensus were selected as candidate compounds. These were six chalcones (cpd26, cpd27, cpd28, cpd31, cpd41, and cpd50), and three flavonoids (DN071_f, IP004, and TP034) with $\Delta G$ values in a range of $-9.0$ to $-10.1$ kcal/mol for YHV 3CL$^{pro}$, and $-8.0$ to $-9.3$ kcal/mol for GAV 3CL$^{pro}$. The 2D structures of these nine compounds were depicted in supplemental Fig. S3, while their 2D interactions were shown in Figs. S4–S5. All screened compounds were stabilized by hydrophobic interactions with YHV 3CL$^{pro}$ residues. The chalcones cpd28, cpd41 and cpd50, and flavonoids IP004 and TP034, can form hydrogen bonding with the imidazole ring of H63, which is one of catalytic residues.

The candidates and the known inhibitor NSC122819 served as primers to generate decoy molecules using the Directory of Useful Decoys, Enhanced (DUD-E) sever (*Mysinger et al., 2012*). The 50 decoys with similar physicochemical properties were created for each compound. All compounds were separately docked into the binding pocket of the YHV 3CL$^{pro}$ model again with all decoys. The receiver operating characteristic (ROC) curve was constructed to validate the docking results (*Empereur-Mot et al., 2015*). In Fig. 1D, the receiver operating characteristic (ROC) curve showed a high true-positive rate over a false-positive rate, with the area under the curve (AUC) of 86.19%. This finding suggested that the molecular docking results were highly predictive and suitable for the viral proteases.

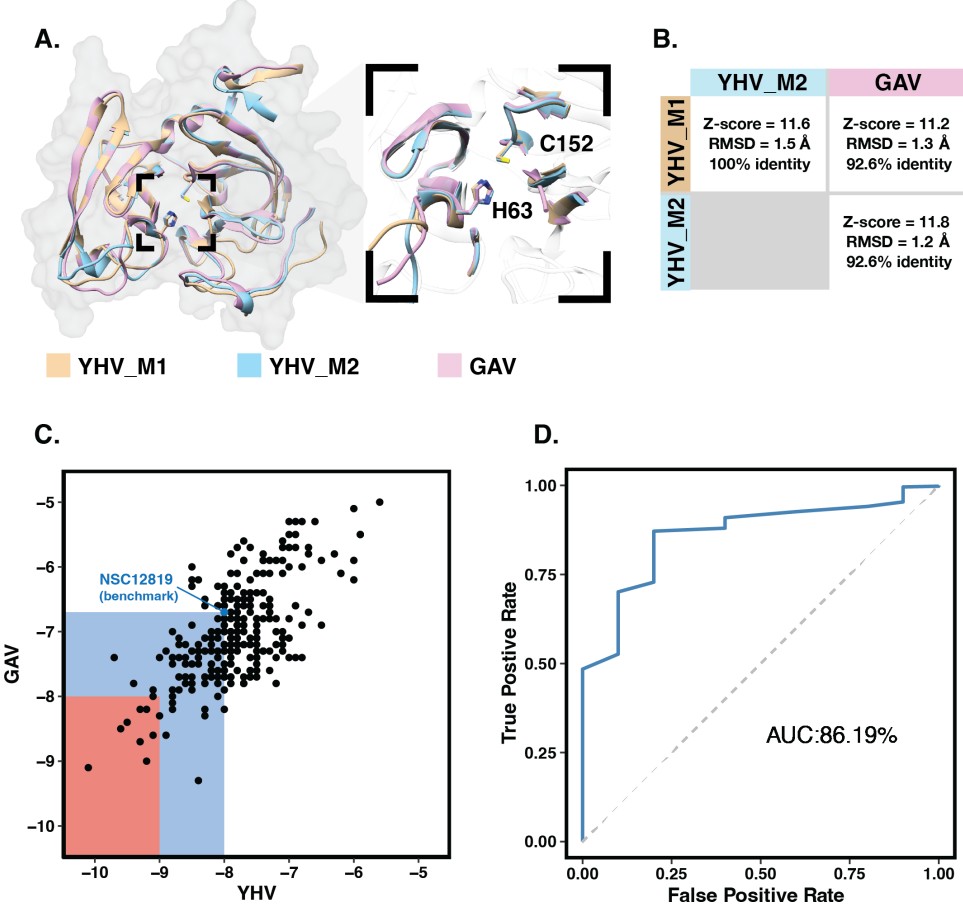

**Figure 1** **Molecular modeling and screening.** (A) Structural comparison of YHV and GAV 3CL$^{pro}$ models with the close-up image of the binding pocket containing the catalytic dyad H63 and C152. (B) Pairwise comparison of Z-score, RMSD, and identity percentage of structural alignment at the binding pocket residues for the models YHV_M1, YHV_M2, and GAV (C) Scatter plot shows binding free energies ($\Delta$G) of the 223 chalcones and 49 flavonoids toward YHV and GAV 3CL$^{pro}$ from molecular docking, where the compounds with $\Delta$G values lower than that of known YHV inhibitor NSC122819 (represented by green dot) are in the green zone, and only the top 15% consensus are in the red zone. (D) Receiver operating characteristic (ROC) curve for active ligands and decoys.

## Drug-likeness prediction

The bioavailability of molecules is one of the most important properties to be considered as a drug. The physiochemical properties of the top nine compounds and NSC122819 were summarized in Table 1. All nine compounds were accepted by Lipinski's of five indicating good physicochemical properties and bioavailability of the molecules. In contrast, the known inhibitor NSC122819 was rejected by Lipinski's criteria with three breaking rules, which indicate that the molecule has poor bioavailability.

BOILED-Egg model result in Fig. 2 shows that most candidates were placed in the yellow and white area, indicating that the molecules were susceptible to gastrointestinal absorption. The flavonoid TP034 and the known inhibitor NSC122819 in the grey area were predicted to be not absorbed in the gastrointestinal tract. Noticeably, the chalcone

**Table 1  Drug-likeness of the 9 candidate compounds and NSC122819.** Calculated physicochemical properties of the candidate compounds compared with known viral inhibitor following Lipinski's rule of five.

| Compound | Lipinski's Rule of Five [60] | | | | | | |
|---|---|---|---|---|---|---|---|
| | MW ($\leq$500 Da) | HBD ($\leq$5) | HBA ($\leq$10) | RB ($\leq$10) | PSA ($\leq$140 Å) | LogP ($\leq$5) | Drug-Likeness |
| cpd26 | 446.45 | 1 | 7 | 7 | 91.29 | 3.38 | Yes |
| cpd27 | 414.45 | 0 | 5 | 9 | 61.83 | 4.15 | Yes |
| cpd28 | 430.45 | 1 | 6 | 9 | 82.06 | 3.95 | Yes |
| cpd31 | 441.48 | 0 | 6 | 9 | 75.47 | 4.16 | Yes |
| cpd41 | 423.48 | 2 | 5 | 7 | 101.08 | 3.13 | Yes |
| cpd50 | 460.59 | 1 | 3 | 7 | 74.86 | 3.64 | Yes |
| DN071_f | 496.49 | 0 | 7 | 5 | 76.33 | 1.15 | Yes |
| IP004 | 421.47 | 1 | 5 | 6 | 90.08 | 3.31 | Yes |
| TP034 | 449.35 | 1 | 11[*] | 7 | 93.99 | 2.61 | Yes |
| NSC122819 | 640.59[*] | 3 | 14[*] | 6 | 173.97[*] | 3.31 | No |

Notes.
   *Lipinski's rule violation. Compound with Lipinski's rule violation <2 is considered as drug-like molecule.

cpd27 could pass through the blood–brain barrier (yellow area) and be pumped out of the brain by activating permeability protein, PGP (*Daina, Michielin & Zoete, 2017*), as suggested by the blue dot.

## MD study of YHV 3CL$^{pro}$ Complexes

The six chalcones and three flavonoids complexed with YHV 3CL $^{pro}$ were investigated by 200-ns MD simulations. The stability of system in term of RMSD was shown in Fig. 3A. The protein structures of most systems including apo protein showed a high RMSD fluctuation with a deviation of 1-2 Å at the early simulation period. The low fluctuation of RMSD values was found after 100 ns until the end of the simulation (<1 Å). The stability of the systems in the last 100 ns is also supported by the RMSD comparison of the MD trajectories in the 2D-RMSD plot (Fig. S6). The ligand-binding at the 3CL$^{pro}$ active site could reduce the motion of the catalytic dyad, governed by a lower RMSD value in complex in comparison to apo-form protein (green line in Fig. 3A).

The number of atom contacts (#Atom contact) within 3.5 Å of the compound at the 3CL$^{pro}$ binding pocket and the distance between a closet heavy atom of ligand and the center of mass of catalytic dyad ($d_{Ligand}$) along with the simulation time were plotted in Figs. S7B–S7C. #Atom contact of each compound was in the same magnitude from the beginning until the end of the simulation, except for a drastically decreased #Atom contact at 170 ns in flavonoid IP004 (Fig. 3B). The drastically change in the interaction of IP004 system was also spotted in distance measurement between the molecule and the catalytic residues, as flavonoid IP004 moved away from the catalytic dyad at 170 ns ($d_{Ligand}$ lengthened from 7.70 to 15.49 Å, Fig. 3C). Result of chalcone cpd41 and known inhibitor NSC122819 also show great distance at the beginning of simulation ($d_{Ligand} = 14.5$ Å and 12.17 Å, respectively). These results of the three molecules indicate their weak interaction to the catalytic dyad. However, the three molecules are still clinging to other residues inside the binding pocket which can still be considered as obstacle for substrate to bind at the

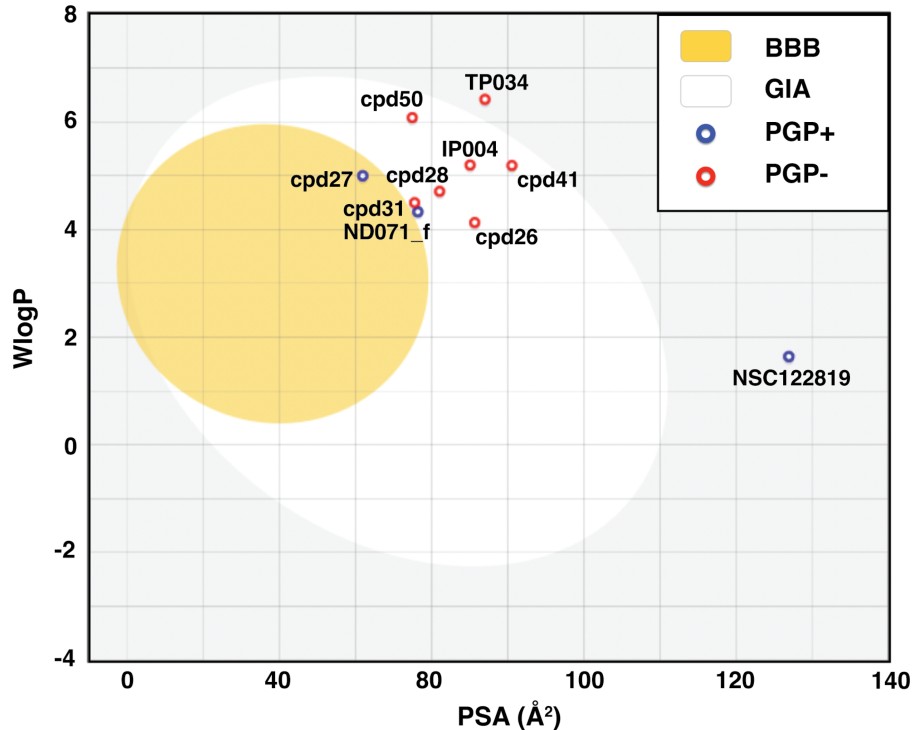

**Figure 2** **Brain Or IntestinaL EstimateD permeation (BOILED-Egg) plot of 9 candidates and reference compound NSC122819.** The blood–brain barrier (BBB) permeable molecules are in the yellow region. The gastrointestinal absorbable (GIA) molecules are in the white area. The blue and red dots indicate permeability glycoprotein substrate (PGP+), and non-permeability glycoprotein substrate (PGP-), respectively.

active site. Other candidates were occupied in the 3CL $^{\mathrm{pro}}$ active site for the whole period of simulation and located closer to the catalytic dyad ($d_{\mathrm{Ligand}}$ <7 Å) than the known inhibitor NSC122819 ($d_{\mathrm{Ligand}} = 12.17$ Å). From the results in Fig. 3, the trajectories between 150 to 200 ns were considered the equilibrated stages and used for analysis in terms of structural dynamics and energetic data.

## Ligand-protein hydrogen bonding

The hydrogen bond is undeniably one crucial factor determining compound binding strength toward the protein. The hydrogen bond formation between compound and 3CL$^{\mathrm{pro}}$ was counted when the distance between hydrogen bond donor (HD) and acceptor (HA) was less than 3.5 Å, and the angle of HD_H…HA was larger than 120° . The number of hydrogen bonding (#Hbond) along with the simulation and the hydrogen bond occupation >10% of selected simulation length (50 ns) were given in Fig. S7. Among the screened compounds, most chalcones formed at least a strong hydrogen bond (>70%) with the backbone of the binding pocket residues, i.e., A172 (cpd26), A180 (cpd41), N194 (cpd26 and cpd31), Y197 (cpd28 and cpd50). Instead, a moderately strong hydrogen bond was detected in the flavonoid DN071_f with the imidazole ring of a catalytic residue, H63 (60.2%). Noticeably, the number of hydrogen bonding formed with IP004 dramatically

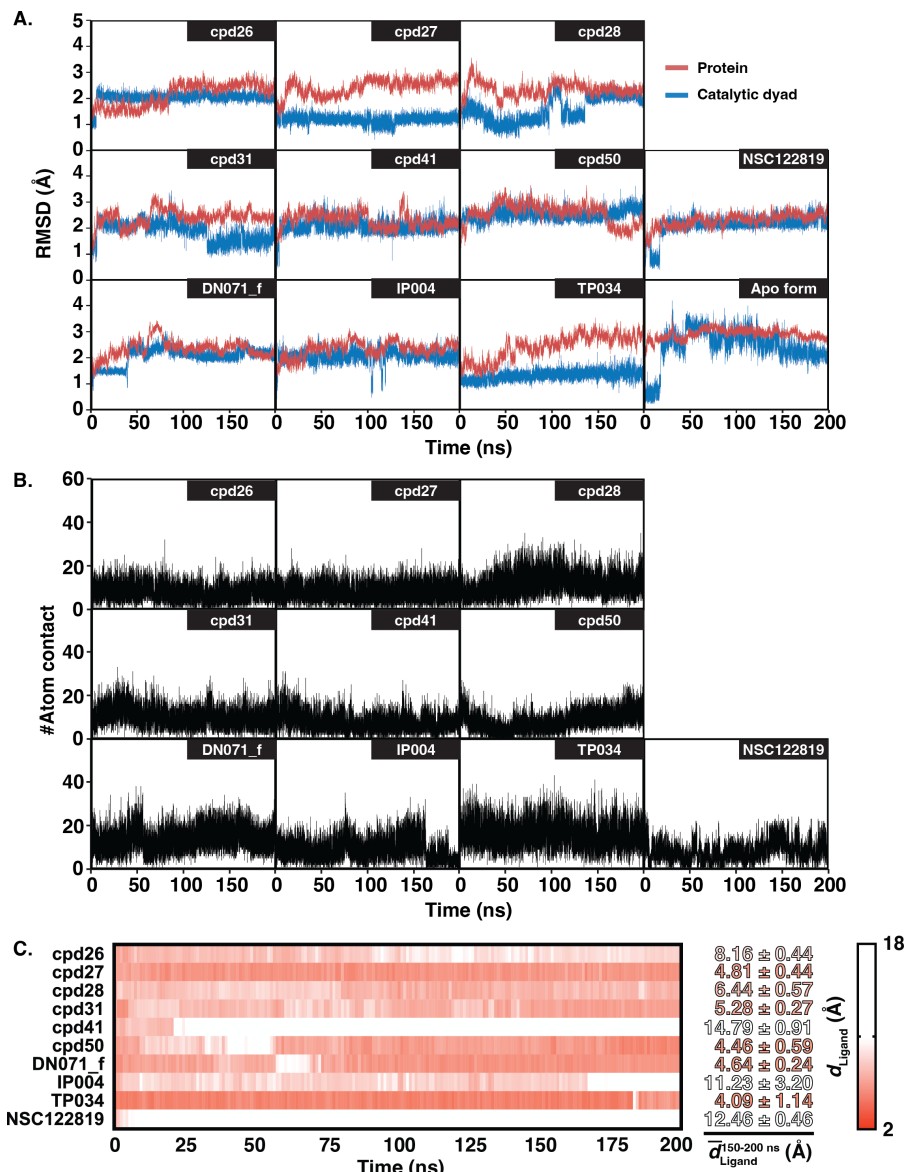

**Figure 3** **RMSD, number of atom contacts, and distance of ligand.** (A) RMSD plot of the alpha car-bons (C$\alpha$) of 3CL $^{pro}$ and catalytic dyad H63 and C152 i n complex and apo systems along with the simu-lation time. (B) Number of atom contacts (#Atom contact) within the 3.5-Åsphere of the compound, and (C) the distance between a closet heavy atom of ligand and center of mass of catalytic dyad ($d_{Ligand}$) for all studied complexes.

declined after 170 ns, which is a result of the same event shown in Figs. 3B–3C as a relatively weak interaction with A180 (29.5%). The chalcone cpd27 and flavonoid TP034 formed the very weak hydrogen bonds with the surrounding residues (<10%).

## Ligand binding affinity

Table 2 showed the $\Delta G_{bind}$ obtained from three end-point binding free energy calculations MM/GBSA, MM/PBSA, and SIE for all nine candidates. The simplified comparison of all

**Table 2   Binding free energies.** Predicted binding free energies ($\Delta G_{bind}$, kcal/mol) of the 9 candidates and the known inhibitor NSC122819 against YHV 3CL$^{pro}$.

| Compound | MM/GBSA | | MMPBSA | | SIE | |
|---|---|---|---|---|---|---|
| | $\Delta G_{bind}$ | SD | $\Delta G_{bind}$ | SD | $\Delta G_{bind}$ | SD |
| cpd26 | −16.3018 | 8.39 | −9.8284 | 9.7856 | −8.81 | 0.51 |
| cpd27 | −16.2456 | 4.7054 | −8.6274 | 5.4369 | −2.47 | 0.26 |
| cpd28 | −6.9181 | 3.2014 | −6.221 | 3.3243 | −7.42 | 0.36 |
| cpd31 | −28.5782 | 3.4971 | −24.9119 | 2.9586 | −9.76 | 0.45 |
| cpd41 | −5.0087 | 3.7453 | −2.2305 | 4.517 | −7.37 | 0.4 |
| cpd50 | −16.7241 | 3.4504 | −12.3697 | 4.5167 | −9.16 | 0.47 |
| DN071_f | −17.8799 | 4.9514 | −16.1626 | 5.2596 | −8.89 | 0.41 |
| IP004 | −9.0483 | 3.8444 | −6.3956 | 6.0151 | −7.18 | 0.41 |
| TP034 | −2.3404 | 5.3616 | −0.9598 | 4.8607 | -7 | 0.39 |
| NSC122819 | −3.044 | 3.9642 | −2.7458 | 5.4544 | −7.6 | 0.4 |

energy calculations was illustrated in the 3D scatter plot (Fig. 4A), which the compounds with better binding affinity were located toward the bottom left corner of the plot. It can be seen that the compounds were found to cluster into three groups corresponding to their predicted $\Delta G_{bind}$ values: (I) the chalcone cpd31 with the best binding energy, lowest $\Delta G_{bind}$ values, predicted by all methods; (II) the chalcones cpd26 and cpd50, and the flavonoid DN071_f with $\Delta G^{MM/PBSA}$ lower than −20 kcal/mol and $\Delta G^{MM/GBSA}$ lower than −25 kcal/mol; and (III) the other screened compounds as well as and the known YHV 3CL$^{pro}$ inhibitor NSC122819 with relatively higher $\Delta G_{bind}$ values, and locate toward the top-right corner of the plot. The compound present in group I and II were considered as best candidates. The simulation systems of the best four compounds, cpd31, cpd26, cpd50, and DN071_f, were repeated in duplicate runs to confirm the reliability of result. The RMSD result of the duplicated simulations show the similar pattern to the original simulation (Fig. S8A). Most of duplications also show the same pattern of hydrogen bonding, except for chalcone cpd26 which the duplicated result show lower number of hydrogen bond at last 50 ns of the simulation (Fig. S8B).

Figure 4B illustrates the residue contribution for ligand binding, where the residues which exhibit $\Delta G_{bind}^{residue}$ lower than −1 kcal/mol were identified as hotspot residues. All candidates from this study interacted with YHV 3CL$^{pro}$ better than the known inhibitor NSC122819 in accordance with the binding affinity results. The hotspot residues were mainly found in the C-terminal domain of 3CL$^{pro}$ (right hand site of the plot), i.e., there were seven to nine stabilized residues for the three chalcones (cpd26, cpd31, and cpd50) and four residues for the flavonoid (DN071_f). Among them, the two nonpolar residues, I168 and V169, and the polar residue Y199 served as the common hotspot residues response in stabilizing all candidates. The chalcones cpd31 and cpd50 and the flavonoid DN071_f also showed interactions with the two hotspot residues on the N-terminal domain, i.e., the catalytic dyad residue H63 and R46 or L49. Only the catalytic residue H63 served as the shared hotspot residue among the 3 compounds. In comparison to cpd26, which
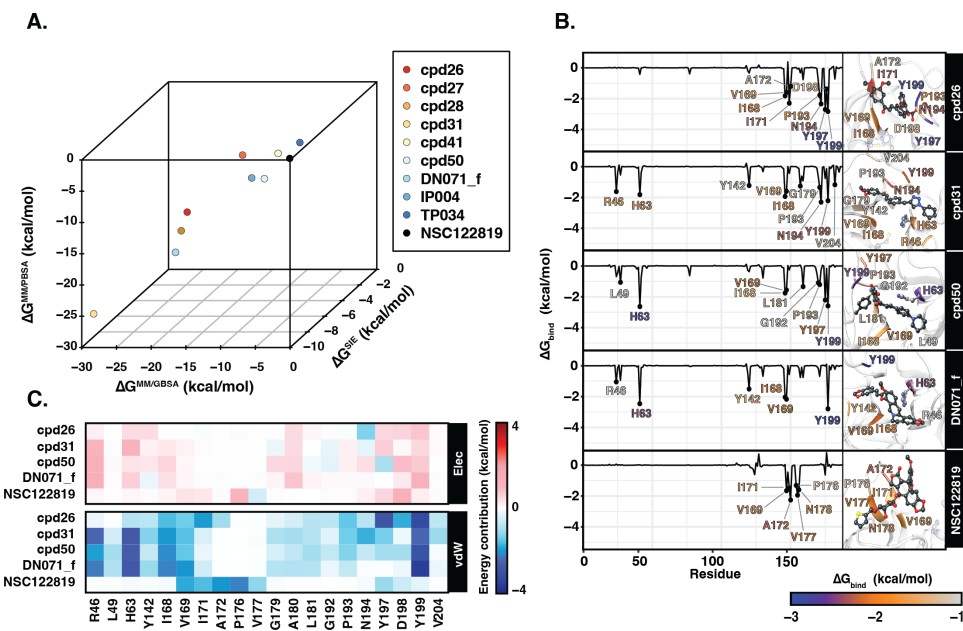

**Figure 4 Binding free energy of 3CL^pro·-ligand complexes.** (A) The 3D scatter plot of binding free energy (ΔG, kcal/mol) predicted by MM/GBSA, MM/PBSA, and SIE methods using the 100 snapshots from the last 50 ns. (B) Per-residue binding free energy decomposition of YHV 3CL^pro bound to the top 4 candidates and the reference molecule NSC122819, where the residues with $\Delta G_{bind}^{residue} < -1.0$ kcal/mol are labeled and colored. The ligands and catalytic residues (H63 and C152) are shown as balls and sticks. (C) Energy contribution of electrostatic and van der Waals energies ($\Delta E_{ele}$ and $\Delta E_{vdW}$) of the key residues contributing to ligand binding. The negative values (high contribution) are represented in more bluish color, while positive values (negative contribution) are represented in more reddish color.

not interact with N-terminal cluster, the compound exhibits the weakest binding affinity among the best four candidates.

In Fig. 4C, the vdW energic term with a negative value in almost all key residues implied that vdW force was the primary interaction for all candidate compounds, similar to the result in the previous study (*Unajak et al., 2014*). The vdW as a major interaction of flavonoids and chalcones toward the binding site of viral 3CL^pro has also been reported in other studies of nidoviruses, especially coronaviruses such as SARs-CoV (*Nguyen et al., 2012*), SARs-CoV-2 (*Das et al., 2021*), and MERS-CoV (*Jo et al., 2019*). This was in concordance with the previous suggestion in the linkage between 3CL^pro of invertebrate nidoviruse, including YHV and GAV, and coronavirus (*Cowley et al., 2000*; *Ziebuhr et al., 2003*). As mentioned earlier, most interactions of candidate compounds in this study are toward nonpolar residue on the C-terminal domain, with only a few interactions toward the catalytic residue on the N-terminal domain. This interaction pattern is similar to the interaction of antiviral compounds against 3C protease (3C^pro) of coxsackievirus and enterovirus (*Sripattaraphan et al., 2022*). Although coxsackievirus and enterovirus 3C^pro employ a catalytic triad, while YHV 3CL^pro uses a catalytic dyad active site, both proteases share several common characteristics since the viruses are identified as members of picornavirus-like supercluster (*Kim et al., 2012*). Although this study primarily focused

on finding anti-viral agents against YHV 3CLpro, due to the fact that YHV and GAV are closely related and GAV 3CLpro was also included in the screening process, all potent candidate compounds present here are potentially provide inhibiting properties for both YHV and GAV 3CLpro.

Searching for YHV anti-viral agent posed many challenges including unavailable of both YHV and GAV viral protein structure data, from either X-ray crystallography, NMR, or cryo-EM experiments. Since YHV and GAV are only two members of their family, Roniviridae, make it difficult to find good reference structures or templates for structural modelling. Moreover, beside the in-silico study, in vitro experiments, such as enzyme-based assay, and in vivo experiments in real animals should be performed to gain better validation of candidates' potential and toxicity. However, the in vitro and in vivo data are not included in this study due to the time and cost constraints in the experiments including newly synthesizing the required amount of potent bioorganic compounds. Further investigation in potential of all candidates presented in this study in both in vitro and in vivo experiments are suggested. The in vitro experiment with the enzyme inhibition assay using potent compounds from this study will be included in our future study.

## CONCLUSIONS

Computational approaches were used to identify potential compounds from in-house database containing chalcones and flavonoids with high predicted bioavailability against YHV 3CL$^{pro}$. As a result, the four potent candidates of three chalcones, cpd26, cpd31, cpd50, and a flavonoid, DN071_f, showed the high binding affinity towards the targeted protease. Most of the interactions were found toward hydrophobic residues on the C-terminal domain of the binding pocket, while some compounds, cpd31, cpd50, and DN071_f, also have interactions directly toward a catalytic residue, H63, on the N-terminal domain. All top candidates with higher binding affinity than reported inhibitor NSC122819 are suggested to be tested by enzyme-based assay for further development as anti-YHV 3CL$^{pro}$inhibitor.

## ACKNOWLEDGEMENTS

The authors would like thank Ponlawoot Raksat, Thitinan Aiebchun, Kamonpan Sanachai, Chonnikan Hanpiboon, Ploylada Vajarintarangoon and other members of Cell and Molecular Biology Laboratory at Faculty of Science Mahidol University and Structural Biology and Biomolecular Modeling at Chulalongkorn University for their kind support.

### Funding

This project was supported by Mahidol University (Fundamental Fund: Basic Research Fund: fiscal year 2022) (Grant no. BRF1-054/2565) and the National Research Council of Thailand, NRCT (Grant number: N42A650231). The funders had no role in study design, data collection and analysis, decision to publish, or preparation of the manuscript.

## Grant Disclosures

The following grant information was disclosed by the authors:

Mahidol University (Fundamental Fund: Basic Research Fund: fiscal year 2022): BRF1-054/2565.

The National Research Council of Thailand, NRCT: N42A650231.

## Competing Interests

The authors declare there are no competing interests.

## Author Contributions

- Kanpong Boonthaworn conceived and designed the experiments, performed the experiments, analyzed the data, prepared figures and/or tables, authored or reviewed drafts of the article, and approved the final draft.
- Kowit Hengphasatporn conceived and designed the experiments, analyzed the data, prepared figures and/or tables, authored or reviewed drafts of the article, and approved the final draft.
- Yasuteru Shigeta conceived and designed the experiments, authored or reviewed drafts of the article, and approved the final draft.
- Warinthorn Chavasiri conceived and designed the experiments, authored or reviewed drafts of the article, provide all the compound structures for screening, and approved the final draft.
- Thanyada Rungrotmongkol conceived and designed the experiments, authored or reviewed drafts of the article, and approved the final draft.
- Puey Ounjai conceived and designed the experiments, authored or reviewed drafts of the article, and approved the final draft.

## Data Deposition

The raw data is available in the Supplemental Files and at Figshare:

The raw data available in supplementary files 6 to 9. The supplementary file 6 contain all 3D structures used in this study, including protein, ligand compounds, known inhibitor, and complexes between 3CLpro and hit compounds. The supplementary file 7 is a table of physicochemical properties of all hit compounds and known inhibitor, used to determine bioavailability and drug-likeness of the compounds. Supplementary file 8 contain all topology files of protein-ligand complexes used in the study. Supplementary file 9 contain topology files of protein-ligand complexes used in duplicated simulation of 4 best compounds and a standard compound.

The other two large supplementary files are uploaded to FigShare as recommended. Link to the files are:

Supplementary file 10, a file contain all trajectories used in calculation.

Trajectories link:

Available at https://figshare.com/s/dcdba4960b816812cfe6

Supplementary file 11, a file contain trajectories of duplicated simulations used in calculation of the best 4 compounds and a standard compound.

Trajectories of duplicated simulations:
Available at https://figshare.com/s/f01ac4b59b9777b10a43

## Supplemental Information

Supplemental information for this article can be found online at http://dx.doi.org/10.7717/peerj.15086#supplemental-information.

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
