# Peer review of "In silico screening of chalcones and flavonoids as potential inhibitors against yellow head virus 3C-like protease"

_PeerJ, doi:10.7717/peerj.15086_

## Round 0.1 · original submission · Major Revisions

Please address concerns of all reviewers and amend your manuscript accordingly.

·

Basic reporting

The manuscript discuss the development of new antiviral agents using structure based design, docking and molecular dynamics targeting specific protease enzyme that is shared between several viruses, so targeting protease enzyme will have impact on the viral infections specially yellow head virus, the method is well known and have been validated by ROC analysis. however; there is some notes
page 8 line 120 should be corrected by giving the reference as the journal style
the discussion is better explain the characters of the biding site, types of interactions,
-provide figures that shows the two dimensional analysis of the protein with the ligand using free Biovia visualizer.
-the language is suitable
-please refer to the following references;
DOI: 10.1007/s11030-022-10434-4
DOI: 10.1002/minf.202200049

Experimental design

The method is well known, there is a typical procedure.
this research is new and interesting regarding the development of broad spectrum antiviral agents
The whole experiments are theoretical without practical wet lab validation.
It is preferred to support the work with enzyme inhibition assay.
the following kit is useful;
https://www.abcam.com/hrv-3c-protease-activity-assay-kit-colorimetric-ab211088.html?gclsrc=aw.ds|aw.ds&gclid=CjwKCAjwp9qZBhBkEiwAsYFsb1xhBaW2TeAkWo3PEL8StKXlPVJ2eDf3zkRs-UhnluCivhK3e6edGxoCVpIQAvD_BwE

Validity of the findings

The validity of the findings could not be confirmed without an enzyme assay, in vitro bioassay could be done using the following kit;
https://www.abcam.com/hrv-3c-protease-activity-assay-kit-colorimetric-ab211088.html?gclsrc=aw.ds|aw.ds&gclid=CjwKCAjwp9qZBhBkEiwAsYFsb1xhBaW2TeAkWo3PEL8StKXlPVJ2eDf3zkRs-UhnluCivhK3e6edGxoCVpIQAvD_BwE
-The results are novel but need in vitro or in vivo validation.
-Conclusion is appropriate.

Additional comments

In vitro, in vivo assay is necessary to confirm the results.
Addition of more related references to support the modeling.

Reviewer 2 ·

Basic reporting

no comment

Experimental design

no comment

Validity of the findings

no cooment

Additional comments

In the study, Boonthaworn et al performed in silico analysis and successfully identified four potent candidate molecules that showed high affinity to YHV and GAV protease. These compounds shows higher binding affinity and higher bioavailability than a compound reported as the best candidate from a previous study. The manuscript is well written and detailed data of the analysis was provided. Minor corrections are listed below.

- L72 in an in vitro -> in vivo and in vitro?
- L296 IP008 -> IP004

Jean-François Picimbon ·

Basic reporting

I find this study well written in part with an interesting and important topic, which is the analysis of therapeutic approaches (inhibitory drugs) to control yellow head virus during prawn cultivation. Accordingly, the authors describe a well established in silico method to check for the interaction of 3C-like-protease with numerous potential inhibitors in a repertoire a 272 chalcones and flavonoids, which could be useful results as 3C-like-protease is a main protease of many various coronaviruses. For these reasons, this type of approach (bioinformatics) is often repeated to analyze protease’s functional binding sites, and perhaps the study lacks a little of originality and needs more generality for PeerJ. This study should be more relevant for a journal of Pharmacology and Bioinformatics or Bioinformatics and Biological Sciences, International Shrimp Farming Prawn, Aquaculture and/or Animals (Special Issue : Advances in Penaeid Shrimp Culture, Feeds and Feedings). In any case, the study needs major revision before further consideration.

Experimental design

1) In Abstract, and Introduction, while the objectives are rather clear, it is lacking the information about the main result: which drug is efficient on the basis of the results?
2) Every protein model needs a reference or template to be built. Please provide the template identity and the percent identity between template and queries (YHV 3CL protease and GAV 3Cl protease).
3) It is not very clear why or how the set of 223 chalcones and 49 flavonoids was chosen.
4) Like 3D modeling and docking, molecular dynamics simulations need to be compared with some specific templates, which need to be defined and described in the Material and Methods section.
5) There are too many algorithms and formula described which are not suitable for PeerJ.
6) Different configurations of the chemical drug should be assessed in docking virtual screening
7) “DeltaG ranged from -9.0 to -10.0 kcal/mol or from -8.0 to -9.3 kcal/mol” is not indicative of high binding affinity. How does it compare with other types of chemicals?

Validity of the findings

7) “DeltaG ranged from -9.0 to -10.0 kcal/mol or from -8.0 to -9.3 kcal/mol” is not indicative of high binding affinity. How does it compare with other types of chemicals?
8) To separate Results and Discussion should be a preferred option. The text would gain clarity and main results or results of interest would appear more clearly. As it stands now, the flow of this section is rather unclear and not compatible with publication in PeerJ.
9) The figures are somehow more explicative, but most of them should be enlarged, the number of figures should be reduced (most of the figures, figure 2, figure 3, figure 5, figure 6 should be used as supplementary figures), and the main point on each figure should be pointed.
10) There should be a table with binding value data.
11) Figure 1B is not pairwise structural alignment of specific binding pocket residues.
12) Figure 8 is poorly legend and lacks clarity.

Additional comments

13) Most of the text under Results and Discussion is matter of Material and Methods.
14) This section should be entirely rewritten focusing on the main results and description of flavonoids and chalcones of interest.
15) Conclusions section should be moved to Abstract and/or final paragraph of Introduction.
16) Conclusion section should be drastically shortened and written in a more cautious way.
Sentence such as “Therefore, we propose that all four candidates, especially for cpd31, could have an inhibitory effect against YHV 3CL and stop the propagation of YHV in real animals” is rather unsuitable for publication in PeerJ.

---

## Round 0.2 · Minor Revisions

Please address remaining concerns of the reviewer and amend your manuscript accordingly

·

Basic reporting

Done

Experimental design

Done

Validity of the findings

Done

Additional comments

the authors have answered all questions critically and made the appropriate modification to the manuscript, the manuscript is now suitable for publication

Jean-François Picimbon ·

Basic reporting

Minor revisions should be required. While I am rather satisfied with the reply to my comments, I find that the reply to reviewer 1's comments needs a little more attention.

Experimental design

"Thank you for your suggestion. In the present study, we aimed to identify potential candidates of
chalcones and flavonoids against the YHV 3CLpro using computational approaches. The computer-aided drug design has been validated for accuracy and reliability enough, as shown in several studies. We have also performed ROC for the computational validation. Moreover, the
result of each step has been compared to the reported compound NSC122819 against bacterially
expressed YHV 3CLpro (Unajak et al. 2014). Due to the time and cost constraints in experimental
study, we have planned to perform the enzyme inhibition assay for future work with this and another series of compounds. In addition, we have to newly synthesize the potent compound, which is another limitation."

The validation of the method should be explained in the experimental design. In addition, the limitations due to the need of bioorganic chemistry before enzyme assay should be mentioned. The lack of specific reference structure should be mentioned as well.

Validity of the findings

"Thank you for your suggestion. In the present study, we aimed to identify potential candidates of
chalcones and flavonoids against the YHV 3CLpro using computational approaches. The computer-aided drug design has been validated for accuracy and reliability enough, as shown in several studies. We have also performed ROC for the computational validation. Moreover, the
result of each step has been compared to the reported compound NSC122819 against bacterially
expressed YHV 3CLpro (Unajak et al. 2014). Due to the time and cost constraints in experimental
study, we have planned to perform the enzyme inhibition assay for future work with this and another series of compounds. In addition, we have to newly synthesize the potent compound, which is another limitation."

These limitations also apply to this section, where the lack of in vivo data should strongly temper the conclusion of the study.

See 108-111: The insights offered from this work can thus open a new avenue to exploit both chalcones and flavonoids in preventive and therapeutic approaches against YHV infection. Since YHV and GAV are closely relative, the structure of GAV is also included in the screening process to obtain the candidate compounds that can be antiviral agents for both pathogens.

To open a new avenue to exploit chalcones and flavonoids in preventive and therapeutic approaches against YHV infection needs 1) in silico data using specific 100% match reference structure or template, not a roughly related model as a template, 2) X-ray or NMR structure, 3) binding assay or spectroscopic data, 4) in vivo data. The limitations of the data presented here should be exposed much more clearly in introduction, material and methods, and discussion sections. Text should be written much more cautiously accordingly (in all sections, including conclusion).

Additional comments

The language is suitable, though English should be once more and thoroughly revised.
See abstract to conclusion: screened? scrutinize? enclosed?………..
were suggested…..are suggested….closely relative….closely related….etc...

---

## Round 0.3 · accepted · Accept

All remaining issues pointed out by the reviewer were adequately addressed and therefore the revised manuscript is acceptable now.